# Application of Ionic Liquids as Mobile Phase Additives for Simultaneous Analysis of Nicotine and Its Metabolite Cotinine in Human Plasma by HPLC–DAD

**DOI:** 10.3390/molecules28041563

**Published:** 2023-02-06

**Authors:** Roxana E. Axente, Miriana Stan, Carmen L. Chitescu, Viorela G. Nitescu, Ana-Maria Vlasceanu, Daniela L. Baconi

**Affiliations:** 1Faculty of Medicine and Pharmacy, Dunarea de Jos University of Galati, 800010 Galati, Romania; 2Department of Toxicology, Carol Davila University of Medicine and Pharmacy, 20021 Bucharest, Romania; 3Ward ATI—Toxicology, Pediatric Clinic 2, Emergency Clinical Hospital for Children Grigore Alexandrescu, 011743 Bucharest, Romania

**Keywords:** ionic liquids, nicotine, cotinine, ion paring, retention factor

## Abstract

Nicotine and cotinine are very polar basic molecules, which makes it difficult to analyze them by reversed-phase liquid chromatography (RPLC), especially in biological samples. Additives with an ionic character have been traditionally used in RPLC as silanol suppressors. The aim of our study was to investigate the potential of selected ionic liquids in improving chromatographic performance in comparison with common additives. The experimental design was conducted using the following ionic liquids as the mobile phase modifiers: 1-butyl-3-methylimidazolium tetrafluoroborate, BMIM[BF_4_] and 1-butyl-3-methylimidazolium hexafluorophosphate BMIM[PF_6_], with a C18 chromatographic column. The separation of these alkaloids on silica-based RPLC stationary phases was successfully conducted by the addition of BMIM[BF_4_] in an acetonitrile:phosphate-buffer-based mobile phase in a pH range of 2.3–5.2. The presented chromatographic method can be used as alternative for monitoring studies or pharmacokinetic application necessary for the evaluation of tobacco smoke exposure.

## 1. Introduction

The use of ionic liquids (ILs) in chromatographic separations has greatly increased since their first use in 1999, as they are considered as ‘‘designer solvents’’ that resolve specific issues, including the substitution of toxic and volatile solvents and the enhancement of separation efficiency or the provision of simplified separations [1,2]. The applications of ILs as mobile-phase additives in liquid chromatography were reported in the last decade and, in many cases, improvements in separation performances have been demonstrated [3,4,5,6]. ILs have been shown to be superior to triethylamine as mobile phase additives in the analysis of a group of β-blockers [5,7]. Improved sensitivity and selectivity and a reduction in analysis time were reported when ILs were tested as mobile phase additives for the HPLC separation of a group of seven fluoroquinolone antibiotics [8]. The addition of ILs was effective and provided a good separation with sharp peaks in the RP–HPLC of five β-Lactam antibiotics [9].

Ionic liquids are salts that are liquid at ambient temperature and are composed only of positively and negatively charged ions, namely organic cations and various anions. Typical cations are asymmetrically substituted nitrogen-containing compounds (imidazolium, pyridinium, quaternary ammonium). Inorganic anions such as Cl^−^, Br^−^, [BF_4_]^−^ and [PF_6_]^−^ serve as ion pairs. Although ILs function as salts, they maintain several types of intermolecular interactions, which are useful for chromatographic separations. ILs have the particularity of providing strong proton-donor–acceptor intermolecular interactions, thus they are able to affect the silanol groups of the silica supports of the commonly used stationary phases in liquid chromatography. It is well known that free silanols cause serious problems in liquid chromatography, particularly in the separation of basic compounds [10]. By adsorbing cations and anions onto the stationary phase, a double layer is created, resulting in hydrophobic, electrostatic and other specific interactions with the stationary phase and solutes that change the retention behavior and the peak shape [6].

While asymmetric cations such as 1-alkyl-3-methyl-imidazolium, 1-alkylpyrrolidinium and N-methyl-N-alkyl-pyrrolidinium react with ionic forms of acidic analytes, large lyophobic anions, containing a high number of fluorine atoms (tetrafluoroborate, hexafluorophosphate), create ion pairs with the protonated forms of the bases [11]. Moreover, the interaction of ILs with the silanol groups of reversed-phase silica-based columns leads to competition between the IL cations and the polar groups of the sorbates of the polar silanol groups on the silica surface, resulting in changes in the retention of the analytes [6,11,12].

Therefore, when diluted for use in a chromatographic mobile phase, ILs may show both synergistic and antagonistic effects in the chromatographic system, providing improvements in the separation selectivity and efficiency [12].

Assessments of nicotine and its metabolites in biological samples are an important component for the evaluation of direct or passive exposure to tobacco smoke. It is now recognized that cotinine, the major metabolite of nicotine, can be used as a biomarker for tobacco addiction, as well as for passive smoking, due to its longer half-life and stability [13,14]. Both nicotine and cotinine can be detected in biological samples using various analytical techniques such as gas chromatography (GC–MS) or high-performance liquid chromatography (HPLC) with a UV [15,16], fluorescence [17] or mass spectrometry detector [18,19]. HPLC methods with UV detectors are simple, rapid, reliable and have a low cost. However, cotinine and nicotine monitoring by liquid chromatography is associated with several inherent analytical difficulties that are partially due to the low concentrations, the similarity of the compound structures and their proprieties [14,19].

Nicotine-related alkaloids belong to a group of very polar basic molecules which, in neutral and acidic environments, always exist in the cationic form [20]. When protonated, basic compounds undergo kinetic ion exchange interactions with the residual silanols on the alkyl-bonded RP columns. This can result in asymmetric peaks, strong retention and low separation efficiency. For this reason, ion pair reagents such as sodium-heptane sulfonate are usually used in order to form a less polar ion pair with a charged solute to achieve a better separation ability [19]. Nicotine is a basic alkaloid with two nitrogen-containing heterocycles, pyridine and pyrrolidine, and it is water soluble and has a pKa of 8.5. Similar to nicotine, cotinine also belongs to the pyrrolidinylpyridines class, but it has a carbonyl group and it is a much weaker base than nicotine with a pKa of 4.8. These differences in the structure and properties of the two compounds can influence their retention behavior and their peak shapes in liquid chromatographic separations that use common mobile phases containing organic solvents and different additives such as acids or phosphate buffers. The separation problems of nicotine and cotinine in liquid chromatography can be successfully solved by using ILs via suppressing the adsorption between the analytes and silanol groups.

In the present study, aqueous solutions of selected ionic liquids were proposed as mobile phase additives for the HPLC analysis of the target compounds nicotine and cotinine in order to solve some analytical difficulties in separation.

Ionic liquids are salts that are liquid at ambient temperature and are composed only of positively and negatively charged ions, namely organic cations and various anions. Typical cations are asymmetrically substituted nitrogen-containing compounds (imidazolium, pyridinium, quaternary ammonium). Inorganic anions such as Cl^−^, Br^−^, [BF_4_]^−^ and [PF_6_]^−^ serve as ion pairs.

The present study aimed to investigate the possibilities of using imidazolium-based ILs as additives for RPLC mobile phases for the effective separation of analytes in the simultaneous analysis of nicotine and its metabolite cotinine in human serum by HPLC with UV–DAD detection. Acetonitrile:water (ACN:H_2_O, 40:60, *v*/*v*) was fixed as the basic mobile phase. Several additives, such as phosphoric acid, phosphate buffer and imidazolium-based ionic liquids, were evaluated in the HPLC analysis of nicotine and cotinine in biological samples. The influence of the anion was evaluated by using ionic liquids with the same cation (1-butyl-3-methylimidazolium) but with different anions such as tetrafluoroborate in BMIM[BF_4_] and hexafluorophosphate in BMIM[PF_6_]. The mobile phase pH and the IL concentration were also investigated regarding the separation efficiency. Retention factors and resolution were examined as parameters, showing an improvement in the separation efficiency.

## 2. Results

### 2.1. Effects of ILs on the Retention Behavior of Nicotine and Cotinine

Several changes can take place in chromatographic systems as a result of the addition of an IL to the mobile phase; imidazolium cations can interact with silanol groups, leading to the suppression of the silanol–cationic solute interaction and can adjust the retention factors of the positively charged analytes [21,22]. On the other hand, anions can be adsorbed onto the hydrophobic stationary phase and can concurrently associate with the cationic analytes [21]. The adsorption affinity of ILs in the stationary phase is influenced by their hydrophobicity and electroneutrality [21].

Therefore, the retention of a basic analyte is a consequence of a mixed mechanism that involves ion pairing, ion exchange and hydrophobic partitioning [5,6]. The synergistic contribution of both the cationic and anionic components of an ionic liquid leads to the very special properties of ILs as mobile phase additives.

During our experiment, it was apparent that the ILs provided a better chromatographic separation even with the addition of the minimum concentration of 0.5 mM in the mobile phase.

The chromatograms for the mixture of nicotine and cotinine, obtained with a LiChrospher^®^ RP-18 column, and the H_2_O:ACN mobile phases with and without BMIM[BF_4_] and BMIM[PF_6_], respectively, are presented in Figure 1 and Figure 2. It was noticed that, in the absence of IL, the separation of nicotine and cotinine was not possible (Figure 1a and Figure 2a), while the addition of both BMIM[BF_4_] and BMIM[PF_6_] clearly improved the separation and the peak shape (Figure 1b and Figure 2c). Similar results were obtained with the phosphate buffer:ACN mobile phases with and without the BMIM[BF_4_] and BMIM[PF_6_]. The chromatograms for the mixture of nicotine and cotinine, obtained with a LiChrospher^®^ RP-18 column, and the phosphate buffer:ACN mobile phases with and without BMIM[BF_4_] and BMIM[PF_6_], respectively, are presented in Figure 3 and Figure 4. The results suggested that BMIM[BF_4_] was superior to BMIM[PF_6_] when added to the mobile phase containing phosphate buffer:ACN in contrast to the behavior of the mobile phase containing H_2_O:ACN.

The identification of each analyte was conducted by a separate injection of the standard solutions.

### 2.2. Effect of the Anion

To determine the effects of the addition of the ionic liquids, the changes in retention of the compounds at different IL concentrations were examined in both the mobile phases considered in the study.

The plots show that the changes in the retention of cotinine with increasing concentration of ILs were rather small with no clear trend (Figure 5). In the phosphate-buffer-based mobile phase, there was a decrease in the retention, while in the water-based mobile phase, there was a constant, but small increase for both the ILs. The maximum retention time for cotinine was 7.8 min in the water-based mobile phase at the maximum concentration of BMIM[PF_6_] of 2.5 mM.

However, the nicotine retention showed a higher increase with the IL concentration in all situations. The maximum retention time for nicotine was 18.8 min in the water-based mobile phase at the maximum concentration of BMIM[PF_6_] of 2.5 mM.

In the presence of BMIM[PF_6_], both analytes showed a higher increase in the retention in both mobile phases (water:ACN and phosphate buffer:ACN), which was consistent with other studies [21,22].

Associated with their short alkyl chains, imidazolium-based cations such as BMIM^+^ or EMIM^+^ paired with a lyotropic anion will increase the retention of cationic basic solutes according to the Hofsmeister series: PF_6_^−^ > BF_4_^−^ [23]. These anions interact with positively charged analytes by a dynamic ion exchange mechanism, leading to an increased retention factor. As the chromatograms show, in our study, the retention times were appreciably longer in the presence of BMIM[PF_6_] compared to those with BMIM[BF_4_], especially for nicotine.

The retention patterns suggested that as long as the adsorbed amount of the IL did not reach the maximal capacity of the column, the interaction of the cationic solutes with the BF_4_^−^ and PF_6_^−^ anions adsorbed onto the stationary phase seemed to be more important than that of the imidazolium cation, and consequently, ion exchange was the main factor responsible for the increased retention.

### 2.3. Effect of Mobile Phase Additives on the Resolution Capability

The addition of the ionic liquids in the mobile phase could have a significant influence on the chromatographic separation. Nicotine and cotinine appeared partially overlapped without the addition of the ionic liquids in both the mobile phases considered in the experiment. The chromatograms showed that the separation of the analytes was significantly improved with the addition of the ILs, even at a very low concentration (0.5 mM) for both the mobile phases. The peaks were completely resolved in the presence of the ILs in all the chromatograms.

Figure 6 shows the variation in the resolution between the target compounds at the different concentrations of the ILs. The results revealed that the optimum condition for the separation of these compounds in a reasonable time was that where the mobile phase consisted of phosphate-buffer:ACN with the addition of [BMIM]BF_4_ at a 1 mM concentration. The resolution improvements can be explained by changes in the selectivity and the relative retentions as well as by the enhancement in the peak profile upon the addition of the ILs.

The peak purity, another useful measurement for global resolution estimation, i.e., the peak area fraction free of interference [24], was checked for the target compound in each chromatogram. Without the ionic liquid, the peak purity was unsatisfactory, with values ranging between 0.031–0.089. In contrast, in the presence of [BMIM]BF_4_, the peak purity ranged between 0.940 and 0.965.

### 2.4. Effect of Mobile Phase pH

As the mobile phase pH greatly impacts chromatographic separation and affects the selectivity, peak shape and retention, in the next step of the experiments, the effect of varying the pH in the aqueous mobile phases containing H_2_O:ACN 60:40 and 1 mM BMIM[BF_4_] and BMIM[PF_6_], respectively, on the chromatographic separation was examined. pH adjustments at 3.5, 5.2 and 9.5 were conducted with a 25% phosphoric acid and ammonia solutions, respectively.

For both the ILs, the separation of the analytes was possible only in the acidic mobile phase (Figure 7 and Figure 8).

A different behavior of the analytes was noted in the presence of the ILs used. Whereas for the mobile phase containing the ion PF6^−^, the retention of the analytes increased linearly with the pH, for the ion BF4^−^, the analytes manifested a decrease in the retention with increasing the pH.

### 2.5. LOD and Linearity

From the mentioned results, the optimum chromatographic conditions were stabilized as follows: 30 mM phosphate buffer:ACN 60:40, containing 1 mM [BMIM][BF4] at pH 2.3 and with a detection at 245 nm.

Under such conditions, five spiked samples in the concentration range of 20–120 ng mL^−1^ for nicotine and 100–450 ng mL^−1^ for cotinine were analyzed for the calibration curves. In the mentioned concentrations range, the target compounds exhibited a good linearity, with R^2^ values of 0.9950 for nicotine and 0.9930 for cotinine.

The LOD (limit of detection) and LOQ (limit of quantification) were determined to be 3.3 times and 10 times, respectively, the standard deviation of the y-intercept divided by the slope of the calibration curve in the matrix sample [25].

The LOD obtained was 16 ng mL^−1^ for nicotine and 48 ng mL^−1^ for cotinine. The LOQ was 48 ng mL^−1^ for nicotine and 145 ng mL^−1^ for cotinine. These values indicated that the method’s sensitivity was appropriate for assessing exposure to nicotine.

## 3. Discussion

The analysis of nicotine and cotinine in plasma and other biological samples is actually of particular interest in monitoring active smoking as well as exposure to tobacco smoke. Various liquid chromatography methods are applied to quantify nicotine and its metabolites, including cotinine in plasma, and the results of such studies have been reported in the last decades. However, the similarity of the two compounds in terms of their structure and properties can result in a number of problems that make their separation difficult using common mobile phases and the most popular chromatographic columns.

Taking into account the need for laboratories to have simple, fast and cheap analytical techniques, as well as the fact that the most technologically advanced techniques do not involve standard laboratory equipment and are too expensive, we proposed a study based on the HPLC method with DAD detection for the analysis of nicotine and cotinine in human plasma.

The main challenge of our study was to provide an improvement in the current chromatographic analysis of polar basic compounds such as nicotine and cotinine that uses common chromatographic columns (i.e., a C18 column) and widely used mobile phases (i.e., acetonitrile–water/phosphate buffer mixtures). This study investigated the use of ILs as mobile phase additives for HPLC in order to improve the separation efficiency for the simultaneous assay of nicotine and its main metabolite cotinine in human plasma. The addition of ILs in the mobile phase for liquid chromatography increases its polarity, resulting in an improvement in the separation mechanism. The reversed-phase stationary phases used in liquid chromatography are based on derivatized silica and have residual silanols on the surface, which interact with basic compounds and produce peak asymmetry and tailing [26,27]. As nicotine and cotinine have basic functional groups, they are able to ionize in the acidic mobile phases, resulting in unfavorable silanol effects. Therefore, the analysis of nicotine and cotinine requires special chromatographic conditions. In the current study, the ILs were selected as the mobile phase additives, as they have advantageous physico-chemical properties (i.e., a lack of vapor pressure and good thermal and chemical stability) as well as silanol-suppressing properties, thus contributing to an improvement in the peak shape and the elimination of peak tailing. It was considered that the selection of appropriate IL additives and their concentration, as well as the optimum pH conditions, would result in an improvement in the separation efficiency and solve the several analytical difficulties determined by the low concentrations of the samples as well as the similarity of the compounds regarding their chemical structures and proprieties.

For basic compounds, even a small amount of silanols in the stationary phase can influence the chromatographic separation and result in decreased retention, peak enlargement and asymmetry. The results of our study showed that the addition of BMIM[BF4] or BMIM[PF6] in low concentrations (i.e., 0.5 mM) to the mobile phase containing water:ACN or phosphate buffer:ACN had a significant beneficial effect on the separation of the two analytes on the C18 stationary phase. The use of imidazolium-based ILs associated with BF_4_^−^ in small amounts (0.5% *v*/*v*) as the mobile phase additives in HPLC has been reported in other studies [28].

A previous study of the chromatographic separation of selected alkaloids using ILs was performed by Petruczynik on a C18 column in eluents containing a mixture of acetonitrile, acetate buffer at pH 3.5 and 2% concentrations of various ionic liquids [29]. A considerable improvement in the peak shape and symmetry and an improved resolution and system efficiency were observed, which was similar to our study. Biogenic amines, β-lactam antibiotics, fangchinoline and tetrandrine were successfully separated whilst using ILs as additives in the mobile phase [9,30,31].

Several concentrations of ILs have been tested in the two different mobile phase. Selectivity and resolution have also been considered. In addition, varying the mobile phase’s pH has been conducted. Using selected ILs, the separation of analytes was only possible in the acidic mobile phase. Regarding the concentration of ILs, usually an increase in the IL concentration in the mobile phase leads to an increase in analyte retention [27]. In our study, this behavior was evident for nicotine, while in case of cotinine, inconsistent results were obtained. The differences between the behavior of nicotine and cotinine can be explained by their structure and physico-chemical proprieties. The acid dissociation constants (pKa) of the analytes in water are 8.5 for nicotine and 4.8 for cotinine. Both are strong bases, which means they were in their cationic state at the working pH of 2.3. In the chemical structure of nicotine there is a pyridine ring (pKa 2.96) and a pyrrolidine ring (pKa 8.5), so that in an acidic aqueous medium, it appears as a diprotonated cation with a charge on both the ionizable fragments [32]. Since in cotinine’s structure there is a conversion of the cyclic amino group into a gamma lactam nitrogen structure, the pyrrolidine ring lost its basicity in this study. Thus, the basicity of cotinine was only due to the protonation of the nitrogen on the pyridine ring. Therefore, one of the important differences between the analytes consisted of their protonation state at the working pH; cotinine was 99% protonated at the single nitrogen atom, while nicotine was rather protonated at both the nitrogen atoms (70%), leading to a stronger interaction with the anions of the ionic liquids, (Marvin.Skecth 19.4 software). However, considering the complexity of the physico-chemical interactions that arise due to the addition of ILs, the causes of the different behaviors of the two analytes can only be presumed.

In discussing the results, the structure, polarity and behavior of the ILs, as well as the properties of the analytes, must be taken into consideration. It is well known that ILs have a dual nature; therefore, both cationic and anionic species may contribute to the chromatographic behavior of analytes.

However, this approach has not been widely used so far, as the knowledge regarding the complex interactions between ILs and chromatographic columns, the components of the mobile phase and the matrix of a real sample is still in an early stage. All the more reason for studies in that direction to be relevant and useful for the development of new methods.

This study contributes to the amount of knowledge regarding HPLC–DAD methods for assays of nicotine and its main metabolite cotinine, providing a new separation method with an elution scheme optimized by using ILs as additives to resolve the both the analytes regarding their appropriate peak shape and resolution.

## 4. Materials and Methods

### 4.1. Chemicals, Reagents and Materials

Organic solvents used were purchased from Merck Romania: acetonitrile, methylene chloride and diethyl ether. Phosphoric acid, potassium dihydrogen phosphate, sodium hydroxide, hydrochloric acid 37%, ammonia 25% and ultrapure water (HPLC-grade), were purchased from Merck Romania. The hydro-organic RPLC mobile phases contained one of the following ionic liquids: 1-butyl-3-methylimidazolium hexafluorophosphate BMIM[PF6] and 1-butyl-3-methylimidazolium tetrafluoroborate BMIM[BF4] (Acros Organics, USA) (Table 1). For the purpose of the experiment, the non-soluble ionic liquid BMIM[PF_6_] was dissolved in a small quantity of about 1 mL of acetonitrile and mixed in the mobile phase.

Analytical standards nicotine and cotinine were purchased form Lipomed GmbH, Weil am Rhein, Germany.

Individual standard stock solutions (1 g mL^−1^) of nicotine and cotinine were prepared in HPLC-grade methanol, and working solutions were prepared daily by appropriate dilution of the stock solution with water. Both the stock solutions and the working solutions were stored at 4 °C.

### 4.2. Analysed Samples

Human plasma obtained from peripheral venous blood was obtained from healthy non-smoking volunteers. All samples were stored frozen (−20 °C) until they were analyzed. Informed consent was obtained from all subjects involved in the study.

Considering the average nicotine and cotinine concentrations reported in studies on smokers’ plasma [13,33], samples of non-smokers’ plasma spiked at 400 ng mL^−1^ for cotinine and 100 ng mL^−1^ for nicotine were analyzed for separation capability testing. To check for interference with endogenous components, blank human samples were analyzed.

### 4.3. Sample Clean-Up and Concentration

A modified method of Massadeh et al., 2009 [19], was used. A 2.5 mL aliquot of plasma was placed into a 10 mL screw-capped glass test tube. The sample was alkalinized with 500 µL of 1 M NaOH and then vortex mixed at 2800 rpm for 30 s (Vortex-Genie 2, Scientific Industries, Bohemia, NY, USA). A 5 mL aliquot of dichloromethane–diethyl ether (1:1 *v*/*v*) was used for one-step single extraction and then vortex mixed at 2800 rpm for 2 min. After centrifugation at 3500× *g* rpm for 10 min (refrigerated centrifuge 2–15 K, Sigma, Darmstadt, Germany), the organic layer was transferred to a new glass tube containing 200 µL of 0.25 M HCl agitated for 30 s and then centrifuged again to separate the phases.

The organic phase was separated and then evaporated under a stream of nitrogen under a high-purity nitrogen flow at 40 °C (Thermo Scientific, Karlsruhe, Germany). The residue was reconstituted in a 500 µL mixture of water:methanol (HPLC-grade) (90:10) and filtrated through a 0.2 µm micro-filter. A 20 µL aliquot was injected into the HPLC and analyzed.

### 4.4. HPLC–DAD Analysis

HPLC analyses were performed with an Agilent 1260 Infinity HPLC–DAD system (Agilent Technologies, Santa Clara, CA, USA). For the separation, an HPLC LiChrospher^®^ RP-18 Column, 5 μm, 4.6 × 250 mm, (Merck, Darmstadt, Germany) was used.

Experimental designs were run with acetonitrile–water phosphate buffer using the ionic liquids BMIM[BF4] and BMIM[PF6] as shown in Table 2.

For the mobile phase H_2_O:ACN 60:40 containing 1.0 mM BMIM[BF_4_] and 1.0 mM BMIM[PF_6_], respectively, pH variation was conducted at 3.5, 5.2 and 9.5 with 25%phosphoric acid or ammonia. The flow rate was set at 1.0 mL/min. The total run time of the method was 30 min. The detection wavelengths were 225 and 245 nm, and the full spectra were recorded over a range of 195–380 nm with a step of 1 nm. Data were evaluated by the OpenLAb ChemStation software from Agilent.

### 4.5. Chromatographic Parameters Estimation

In this work, the retention factor, k, and the resolution, R, were calculated according to well-established equations.

## 5. Conclusions

The mobile phases containing the ionic liquids 1-butyl-3-methylimidazolium hexafluorophosphate BMIM[PF6] and 1-butyl-3-methylimidazolium tetrafluoroborate BMIM[BF4] as additives were successfully applied for the analysis of nicotine and cotinine in human plasma by HPLC with DAD detection in an RP-C18 column.

In the mobile phases without the addition of the ILs, the peaks of nicotine and cotinine appeared overlapped, and the separation efficiency was low. The addition of ILs in the mobile phase increased the complexity of the chromatographic system due to the interaction of both the cations and anions with the stationary phase and due to the interaction of the anions with the cationic analytes, leading to an ion-pairing effect in the mobile phase.

This work showed a significant improvement in the chromatographic analysis of the polar compounds nicotine and cotinine with acetonitrile–water/phosphate buffer mixtures by the addition of ionic liquids, especially BMIM[BF4] at a concentration of 1 mM in acidic conditions.

The study contributes to the analytical methodology for nicotine and cotinine assays in biological samples and offers perspectives for the use of an improved HPLC method in terms of analyte separation and peak resolution. The simultaneous analysis of nicotine and cotinine is required in biological samples for monitoring of tobacco exposure, and separation efficiency in the applied HPLC methods is essential.

Therefore, the presented chromatographic method can be used as alternative for monitoring studies or the pharmacokinetic applications necessary for the evaluation of tobacco smoke exposure.

## Figures and Tables

**Figure 1 molecules-28-01563-f001:**
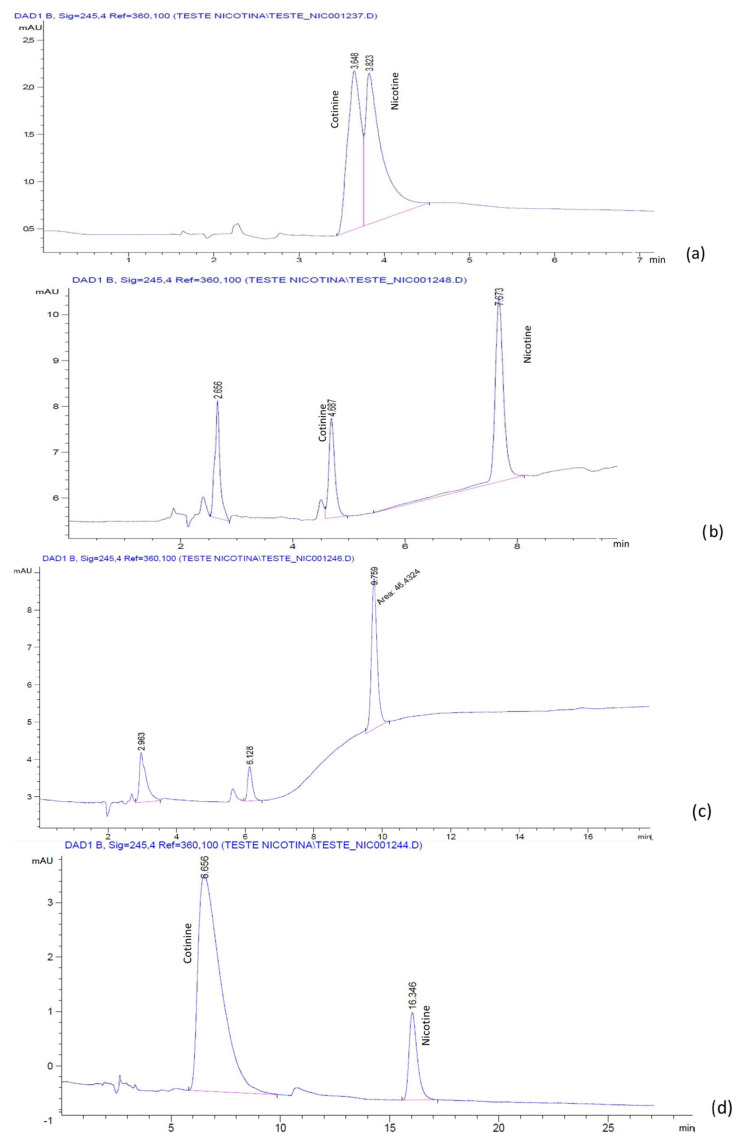
Chromatograms of nicotine and cotinine in human plasma with mobile phases containing H_2_O:ACN 60:40 (pH 2.5 with H_3_PO_4_): (**a**) without additive; (**b**) with 0.5 mM BMIM[BF_4_]; (**c**) with 1.0 mM BMIM[BF_4_]; (**d**) with 2.5 mM BMIM[BF_4_].

**Figure 2 molecules-28-01563-f002:**
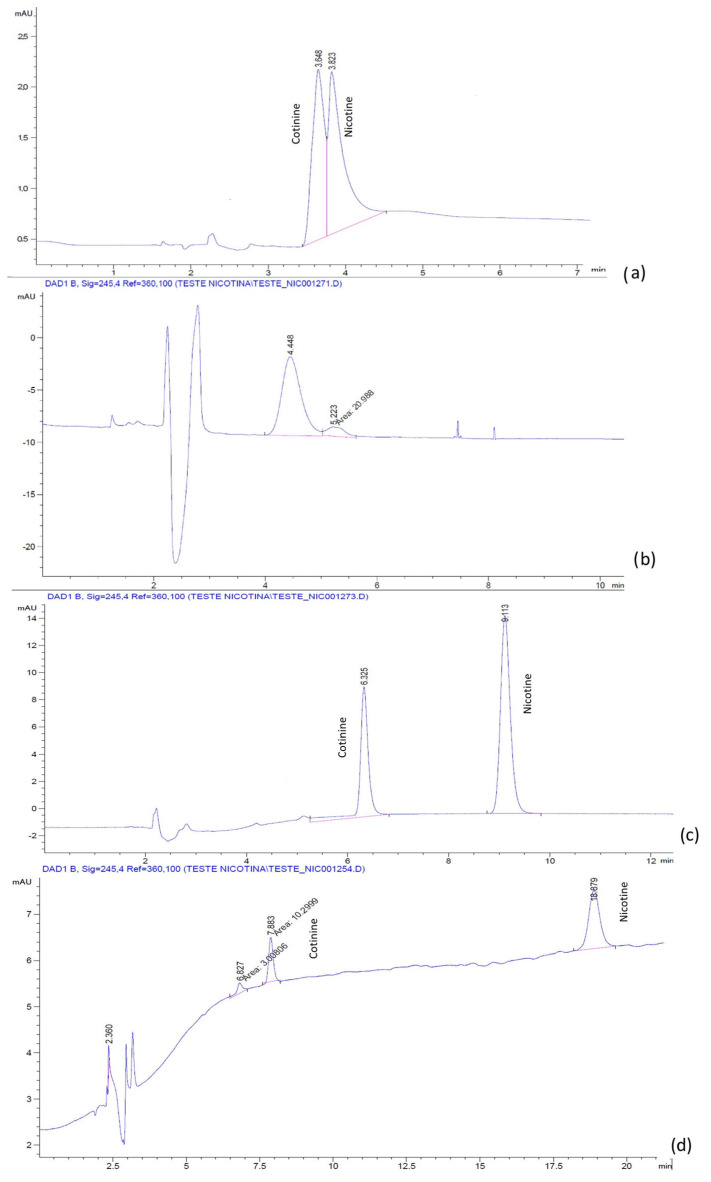
Chromatograms of nicotine and cotinine in human plasma with mobile phases containing H_2_O:ACN 60:40 (pH 2.5 with H_3_PO_4_): (**a**) without additive; (**b**) with 0.5 mM BMIM[PF_6_]; (**c**) with 1.0 mM BMIM[PF_6_]; (**d**) with 2.5 mM BMIM[PF_6_].

**Figure 3 molecules-28-01563-f003:**
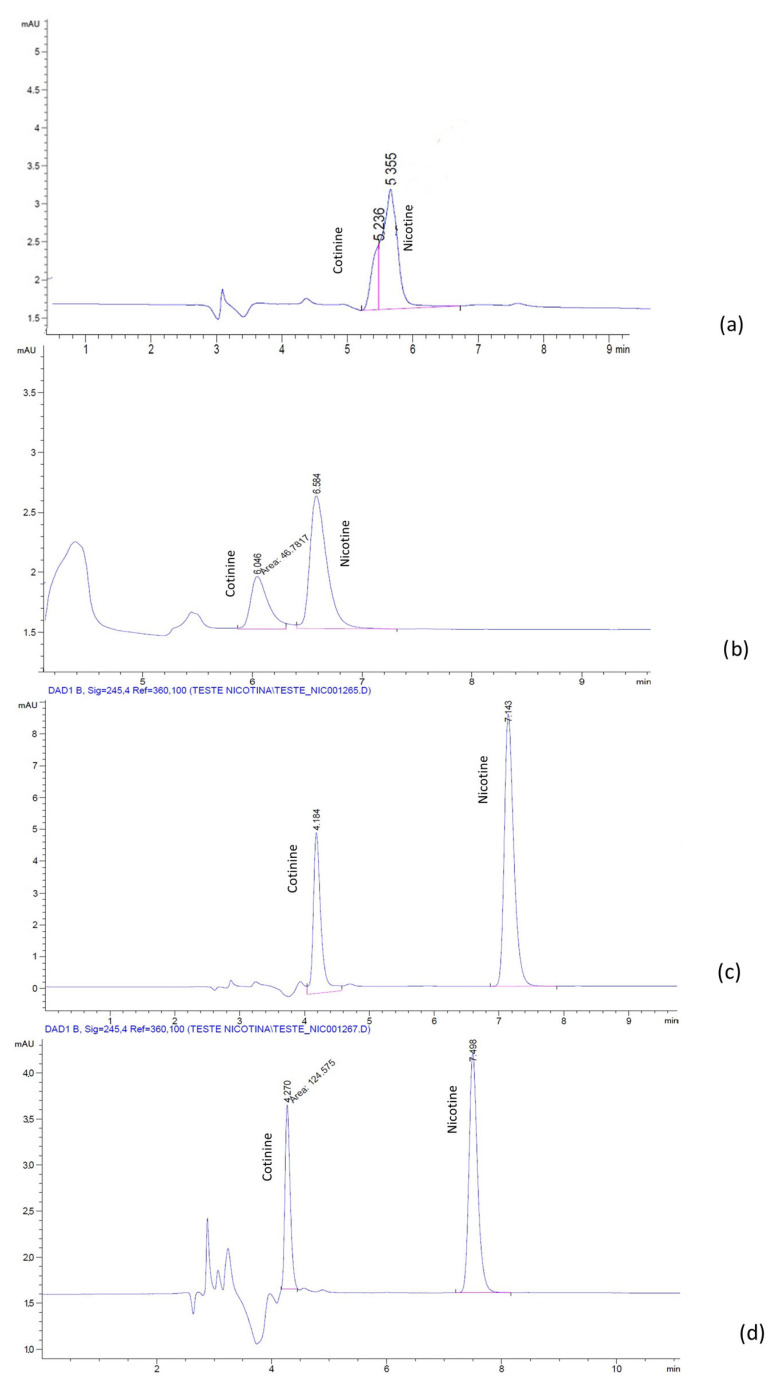
Chromatograms of nicotine and cotinine in human plasma with mobile phases containing phosphate buffer:ACN 60:40 (pH 2.3 with H_3_PO_4_): (**a**) without additive; (**b**) with 0.5 mM BMIM[BF_4_]; (**c**) with 1.0 mM BMIM[BF_4_]; (**d**) with 2.5 mM BMIM[BF_4_].

**Figure 4 molecules-28-01563-f004:**
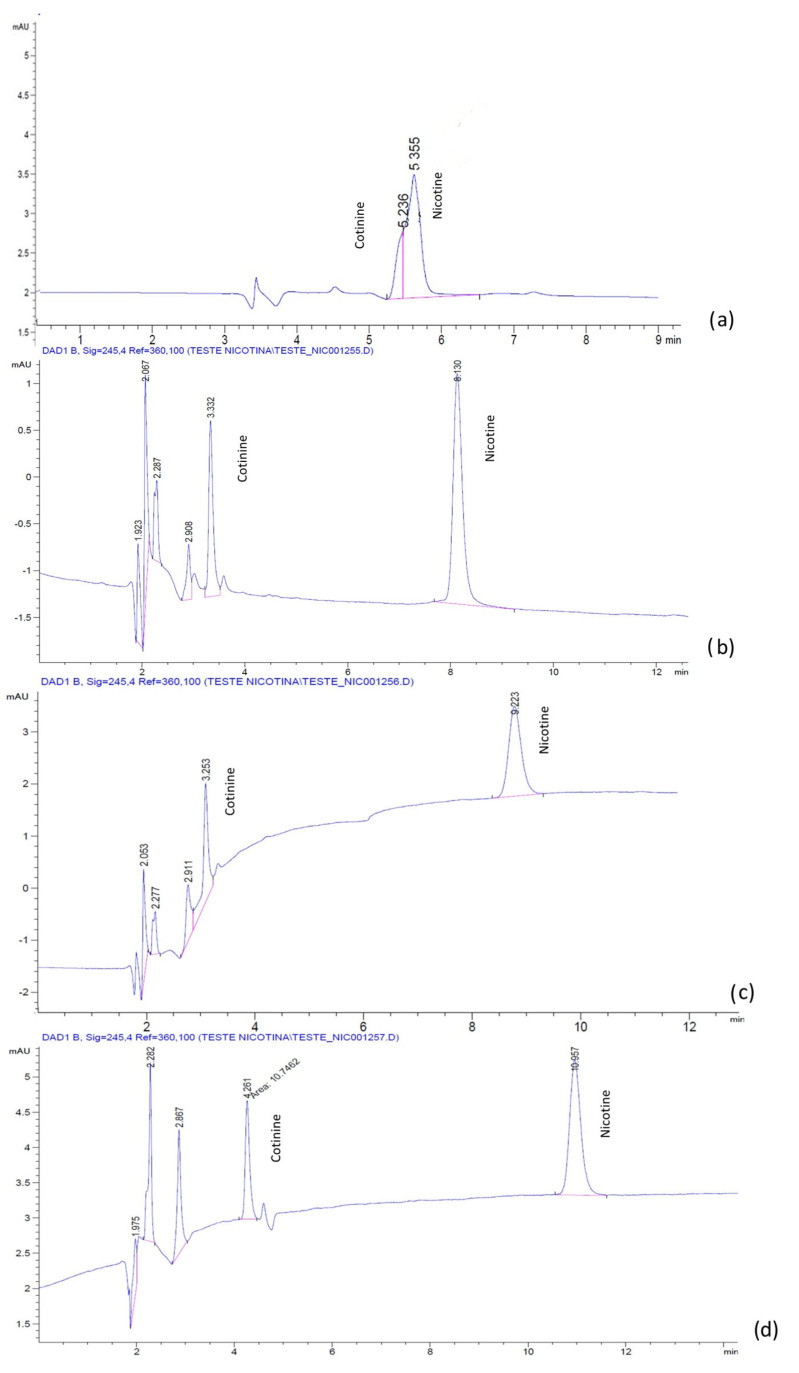
Chromatograms of nicotine and cotinine in human plasma with mobile phases containing phosphate buffer:ACN 60:40 (pH 2.3 with H_3_PO_4_): (**a**) without additive; (**b**) with 0.5 mM BMIM[PF_6_]; (**c**) with 1.0 mM BMIM[PF_6_]; (**d**) with 2.5 mM BMIM[PF_6_].

**Figure 5 molecules-28-01563-f005:**
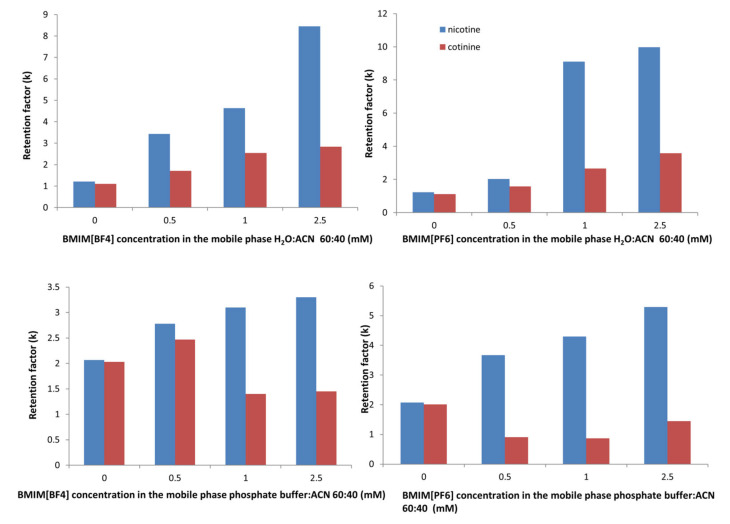
Effect of the concentration of ILs in the mobile phases on the retention factor, k, of the analytes.

**Figure 6 molecules-28-01563-f006:**
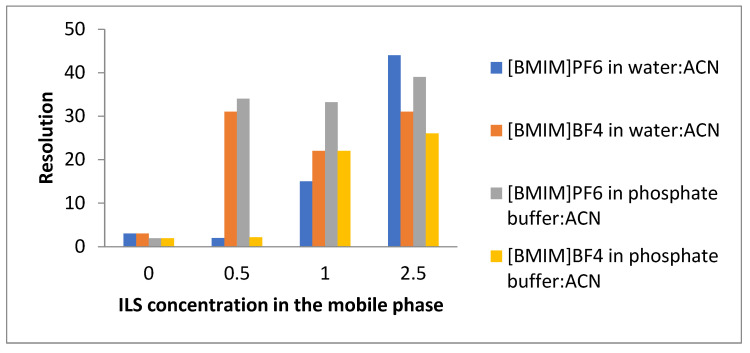
Effect of IL concentration on the resolution of the target compounds by using different mobile phases.

**Figure 7 molecules-28-01563-f007:**
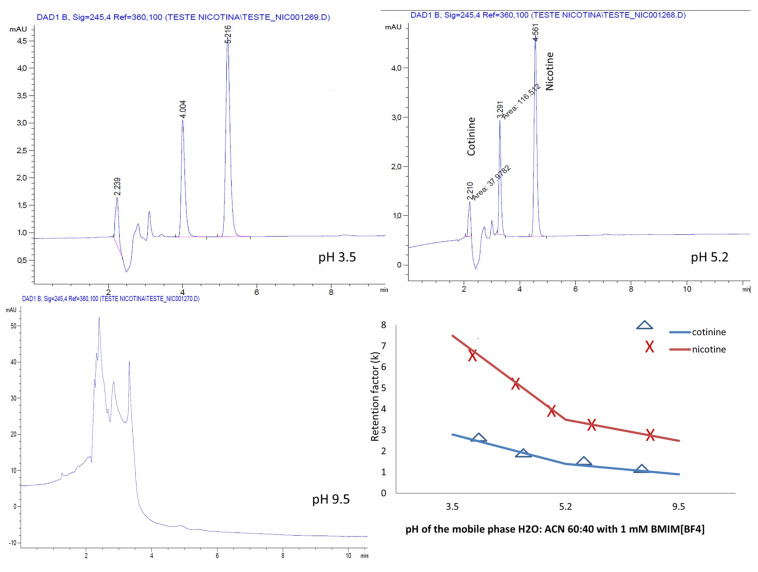
Chromatograms and variation in the pH-dependent retention factor for nicotine and cotinine with mobile phases containing H_2_O:ACN 60:40 and 1 mM BMIM[BF_4_].

**Figure 8 molecules-28-01563-f008:**
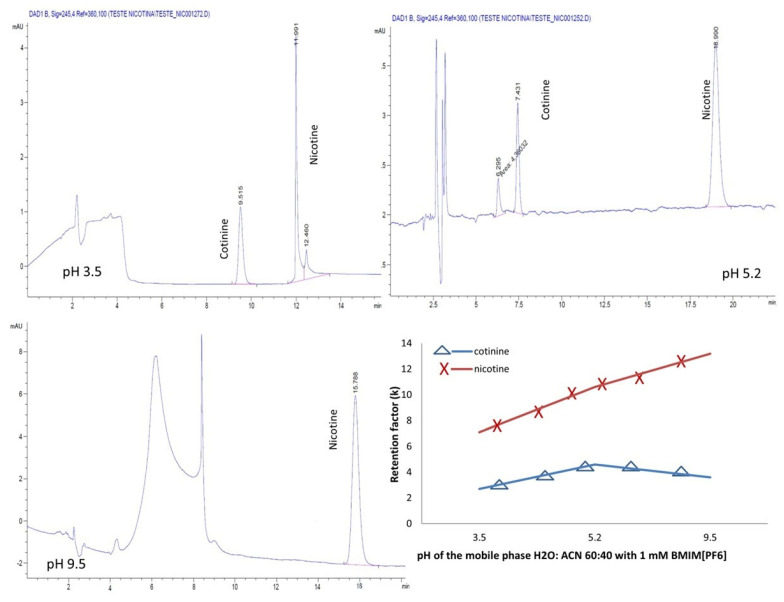
Chromatograms and variation pH-dependent retention factor for nicotine and cotinine with mobile phases containing H_2_O:ACN 60:40 and 1 mM BMIM[PF_6_].

**Table 1 molecules-28-01563-t001:** Structure and properties of the ionic liquids used in the presented study.

Ionic Liquid	Cation	Anion	Density (g/mL)	m.p. (°C)	Water Solubility
BMIM[PF_6_]	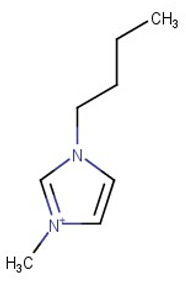 1-Butyl-3-methylimidazolium	PF_6_^−^ 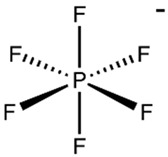	1.21	11	non soluble
BMIM[BF_4_]		BF_4_^−^ 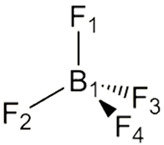	1.31	−71	Soluble

**Table 2 molecules-28-01563-t002:** Experimental design regarding the mobile phase used.

1. H_2_O:ACN 60:40, pH 2.5 with H_3_PO_4_	No additive
	BMIM[BF4] concentrations of 0.5 mM, 1.0 mM and 2.5 mM
	BMIM[PF6] concentrations of 0.5 mM, 1.0 mM and 2.5 mM
2. 30 mM phosphate buffer KH_2_PO_4_:ACN 60:40, pH 2.3 with H_3_PO_4_	No additive
	BMIM[BF4] concentrations of 0.5 mM, 1.0 mM and 2.5 mM
	BMIM[PF6] concentrations of 0.5 mM, 1.0 mM and 2.5 mM

## Data Availability

The data presented in this study are available on request from the corresponding author.

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
