# Peer review of "Application of Ionic Liquids as Mobile Phase Additives for Simultaneous Analysis of Nicotine and Its Metabolite Cotinine in Human Plasma by HPLC–DAD"

_molecules, 2023, doi:10.3390/molecules28041563_

Round 1

Reviewer 1 Report

Manuscript ID= molecules-2167122

After carefully evaluation. I am pleased to send you some comments. Please consider these suggestions as listed below to revise the article .  

  1. The title seems ok.
  2. The abstract seems to be good. Please add one more introductory line of your objective in beginning of abstract.
  3. Please give maximum 4-5 most suitable keywords.
  4. Research gap should be delivered on more clear way with directed necessity for the future research work.
  5. Introduction section must be written on more quality way, i.e., more up-to-date references addressed.
  6. The novelty of the work must be clearly addressed and discussed, compare previous research with existing research findings and highlight novelty.
  7. What is the main challenge?
  8. Please check the abbreviations of words throughout the article. All should be consistent.
  9. What is problem statement?
  10. The main objective of the work must be written on the clearer and more concise way at the end of introduction section.
  11. Please provide space between number and units. Please revise your paper accordingly since some issue occurs on several spots in the paper.
  12. The methodology part is fine. NO comment.
  13. Please highlights the peaks in figure 1. Also it’s not well scientifically explained.
  14. The entire result section there is no scientific support for your result why?
  15. The results does not have any scientific discussion.
  16. Why author explain the results in a very general way.
  17. Please include all chemical/instrumentation brand name and other important specification.
  18. Please provide high quality image for figure 1 and 2.
  19. Regarding the replications, authors confirmed that replications of experiment were carried out. However, these results are not shown in the manuscript, how many replicated were carried out by experiment? Results seem to be related to a unique experiment. Please, clarify whether the results of this document are from a single experiment or from an average resulting from replications. If replicated were carried out, the use of average data is required as well as the standard deviation in the results and figures shown throughout the manuscript. In case of showing only one replicate explain why only one is shown and include the standard deviations.
  20. Please add a comparative discussion section.
  21. Conclusion and Future perspectives should be revised carefully. Conclusion section is missing some perspective related to the future research work, quantify main research findings, and highlight relevance of the work with respect to the field aspect.
  22. To avoid grammar and linguistic mistakes, MAJOR level English language should be thoroughly checked. Please revise your paper accordingly since several language issue occurs on several spots in the paper.
  23. Reference formatting need carefully revision. All must be consistent in one format. Please follow the journal guidelines.  The table and figure formatting also need carefully revision. The formatting is strange.

Decision = Encourage you to revise after MAJOR revision. It was tough for me to read and go through, but I was able to make a few remarks. Please put forth your best efforts and resubmit revised version.

Reviewer 2 Report

Row 18 "peak resolution and was investigated

Row 23 "solved by the addition of of BMIM[BF4] in an"

Row 106, Figure 2 "H2O:CAN 60:40" 

Why you choose different pH values for he mobile phase?

Reviewer 3 Report

The manuscript presents the effect of the addition of ionic liquid on the retention and resolution of nicotine and cotinine. The idea of the work is interesting. Unfortunately, the execution is very poor. I cannot recommend this work for publication.

First, I do not understand why Authors used human plasma samples for method optimization. It is much better to check the ILs effect for standard samples and apply the chosen conditions for real samples. 

 What was the reason for the changes in the shape of the baseline? It should be flat in isocratic elution.

The quality of figures 1, 2, 3, 4, 7, and 8 is extremely poor. The chromatograms are of poor resolution, lack scale, and are illegible.

The relationship presented in Figure 5 and 6 certainly has no kinks at an angle. Please consider presenting the data.

pKa is not a dissociation constant

Giving equations for retention factor and resolution makes no sense. 

Round 2

Reviewer 1 Report

I re-review again the revised version. It seems ok to me but still there are several gramatical error in each section. I recommend again a English revision. Second the resolution of images are still poor. It should be high to publish. Please provide it again.

Reviewer 3 Report

The manuscript was corrected according to suggestions, and a valuable response was provided. I can accept current version.